# 13q Deletion Syndrome Presenting with Lymphopenia Detected Through Newborn Screening for Primary Immunodeficiencies

**DOI:** 10.3390/ijms26199302

**Published:** 2025-09-23

**Authors:** Irina Efimova, Anna Mukhina, Zhanna Markova, Sergey Mordanov, Irina Soprunova, Dmitry Pershin, Natalya Balinova, Yunna Petrusenko, Dmitry Meleshko, Rena Zinchenko, Nadezhda Shilova, Sergey Voronin, Anna Shcherbina, Sergey Kutsev, Andrey Marakhonov

**Affiliations:** 1Research Centre for Medical Genetics, Moscow 115522, Russia; ffmanya@yandex.ru (A.M.); balinovs@mail.ru (N.B.); renazinchenko@mail.ru (R.Z.); nvsh05@mail.ru (N.S.); voronin.sv@med-gen.ru (S.V.); kutsev@mail.ru (S.K.); 2Dmitry Rogachev National Medical Research Center of Pediatric Hematology, Oncology, and Immunology, Moscow 117198, Russia; dimprsh@icloud.com (D.P.); shcher26@hotmail.com (A.S.); 3Medical Genetics Laboratory, Rostov State Medical University, Rostov-on-Don 344022, Russia; labmed@mail.ru; 4Center for Family Health and Reproduction, Astrakhan 414040, Russia; irinkagen16@yahoo.com; 5“Biotech Campus” LLC, Moscow 117437, Russia; ypetrusenko@biotc.ru (Y.P.); dmeleshko@biotc.ru (D.M.); 6Shemyakin-Ovchinnikov Institute of Bioorganic Chemistry, Russian Academy of Sciences, Moscow 117997, Russia; 7Principal Engineering School, ITMO University, Saint-Petersburg 191002, Russia

**Keywords:** 13q deletion syndrome, newborn screening, primary immunodeficiency, TREC/KREC assay, long-read sequencing

## Abstract

The expanded newborn screening (NBS) program in the Russian Federation, launched in 2023, includes the detection of severe forms of T- and B-cell immunodeficiencies via TREC/KREC quantification. We report a rare case of a male infant having multiple congenital anomalies and lymphopenia identified through this program. Genetic testing revealed a 25.8 Mb terminal deletion spanning 13q31.2–qter, consistent with 13q deletion syndrome. Initial NBS revealed reduced TREC levels, prompting further evaluation. The patient exhibited a complex phenotype, including central nervous system malformation (alobar holoprosencephaly), severe congenital heart disease, renal hypoplasia, limb and genitourinary anomalies, and facial dysmorphism. Postnatal complications included pneumonia, pleuritis, and chylothorax. Flow cytometry demonstrated mild T- and B-cell lymphopenia. The genomic defect was characterized using long-read third-generation sequencing, enabling precise breakpoint identification and accurate mapping of deleted genes. The deletion was confirmed via subtelomeric FISH analysis. The patient died at 7 months of age due to the progression of underlying congenital anomalies and associated complications. Our findings broaden the clinical characterization of distal 13q deletion syndrome and demonstrate the value of long-read sequencing in structural chromosomal analysis. They further highlight the difficulties of caring for neonates having complex malformations and immune dysfunction. Given the potential for both primary and secondary immune disturbances, comprehensive immunological evaluation should be considered in patients having 13q deletion syndrome to improve diagnostic accuracy and inform appropriate clinical management.

## 1. Introduction

Inborn errors of immunity (IEI), also known as primary immunodeficiencies (PIDs), are a heterogeneous group of disorders characterized by susceptibility to infections, malignancies, immune dysregulation, inflammation, and other clinical manifestations [1]. As of 2024, more than 550 genetically distinct conditions have been identified [2], and this number continues to grow rapidly.

Newborn screening (NBS) is a vital tool for the early detection of severe forms of T- and B-cell immunodeficiencies, including severe combined immunodeficiency (SCID)—the most severe form of T-cell lymphopenia. Modern NBS programs utilize the quantification of T-cell receptor excision circles (TRECs) and kappa-deleting recombination excision circles (KRECs) in neonatal dried blood spots (DBSs) [3]. The TREC assay is specifically designed to detect T-cell lymphopenia, including SCID, while KREC quantification enables the identification of congenital agammaglobulinemia, a disorder characterized by severe B-cell deficiency [4,5,6,7].

Although the primary goal of TREC/KREC-based NBS is to identify infants having SCID and other severe forms of T- and/or B-cell deficiencies, it has also proven effective in detecting a wide range of syndromic conditions [8,9,10]. In many of these cases, immune abnormalities are present—often manifesting as mild or transient lymphopenia or abnormal TREC/KREC values—that do not fully correspond to the established IEI categories [1,11,12].

Since 2023, Russia has implemented a nationwide newborn screening program for SCID and agammaglobulinemia, using the TREC/KREC assay [13]. Beyond the expected cases of IEI, this screening has identified newborns having syndromic features and previously unrecognized immune abnormalities, expanding our understanding of immune dysregulation in various genetic syndromes.

Here, we report a case of a patient with 13q deletion syndrome presenting with immunological abnormalities, highlighting the broader diagnostic potential of NBS.

## 2. Materials and Methods

The NBS program was carried out following the methodology described elsewhere [14]. Informed written consent was obtained from parents before enrolling their newborns in the program. Dried blood spot (DBS) cards were collected on the second day of life for full-term infants and on the seventh day for preterm infants and were then sent to one of 10 Centers for Expanded NBS for first-tier PCR.

The first-tier PCR was performed in these centers to assess TREC, KREC, and the homozygous deletion of exon 7 in the *SMN1* gene using the NeoScreen SMA/TREC/KREC assay (DNA-Technology Ltd., Moscow, Russia) according to the manufacturer’s recommendations.

Newborns whose first-tier PCR results were below the cut-off were referred to the Research Centre for Medical Genetics (RCMG) for second-tier PCR testing using the Eonis™ SCID-SMA kit (Wallac Oy, Turku, Finland), following updated protocols [12]. According to the revised algorithm, a TREC value below 150 copies per 10^5^ cells was considered an indication for further diagnostic evaluation. In addition, TREC values in the range of 150–200 copies per 10^5^ cells were regarded as a high-risk zone: newborns within this range were referred for follow-up if a syndromic phenotype was present. In such cases, confirmatory diagnostics—including immunophenotyping and genetic testing—were initiated to clarify the diagnosis and guide clinical management [14].

Subsequent immunological confirmation of the IEI diagnosis was performed according to the lyse-no-wash manufacturer’s protocol (Beckman Coulter, Indianapolis, IN, USA) for the multi-color flow cytometry method, using a Beckman Coulter CytoFLEX flow cytometer and a custom dry format DURA Innovations antibody panel (LUCID product line, Beckman Coulter, USA) at the Dmitry Rogachev National Medical Research Center of Pediatric Hematology, Oncology and Immunology (Moscow, Russia), as described earlier [15]. Reference values were used as determined elsewhere [16].

Long-read third-generation sequencing was performed using the Oxford Nanopore Technologies PromethION platform in a framework of the National Genetic Initiative “100,000+Me”. High molecular weight (HMW) genomic DNA was extracted from peripheral blood leukocytes using the Monarch^®^ HMW DNA Extraction Kit for Blood. Concentration and purity were determined by spectrophotometry using Qubit dsDNA HS assay with NanoDrop (Thermo Fisher, Waltham, MA, USA). Size distribution and integrity were evaluated by pulsed-field gel electrophoresis (PFGE) on Femto Pulse (Agilent Technologies, Santa Clara, CA, USA). Sequencing libraries were prepared using the Oxford Nanopore Technologies (ONT) Ligation Sequencing Kit (SQK-LSK114; https://nanoporetech.com/document/genomic-dna-by-ligation-sqk-lsk114, assessed on 3 December 2024) following the manufacturer’s protocol for unsheared HMW DNA. All bead cleanups used room-temperature reagents, gentle pipetting with wide-bore tips, and extended elution to maximize recovery of long molecules. For each flow cell load, 200–300 fmol of adapter-ligated DNA was prepared in a sequencing buffer (SQB) with loading beads (LB) immediately before loading. Libraries were sequenced on a PromethION 48 using R10.4.1 flow cells (FLO-PRO114M) and MinKNOW control software (version 25.05) (Oxford Nanopore Technologies, Oxford, UK).

Interphase FISH on preparations from the patient’s uncultured peripheral blood lymphocytes was carried out using a DNA probe specific to the subtelomeric region of the long arm of chromosome 13 (Sub Telomere 13qter; KREATECH, Amsterdam, The Netherlands) following the manufacturer’s protocol. The FISH results were analyzed using an AxioImager M.1 epifluorescence microscope (Carl Zeiss, Jena, Germany) and Isis digital image processing software (version 9.0.0) (MetaSystems, Altlussheim, Germany).

## 3. Clinical Report

The male patient was born to unrelated parents having an unremarkable family history. The older sibling is healthy. Ultrasonography in the first trimester revealed a high risk of fetal growth restriction (FGR) before 37 weeks (1 in 552). In addition, multiple congenital anomalies were identified in the fetus. Due to the high risk of chromosomal abnormalities, prenatal karyotyping of the fetus was recommended but was not performed due to maternal refusal. Follow-up ultrasonography confirmed FGR. A perinatal consultation at 19 weeks and 4 days of pregnancy concluded a diagnosis of FGR and multiple congenital anomalies, including malformations of the central nervous system, heart, and genitourinary tract, as well as phenotypic features suggestive of a high risk for chromosomal abnormalities. In the third trimester, at 29 weeks of gestation, fetal assessment confirmed FGR and multiple congenital anomalies, including alobar holoprosencephaly, a functionally single left ventricle with tricuspid valve atresia, double outlet of the great vessels with severe pulmonary artery hypoplasia/atresia, and renal hypoplasia. A chronological summary of the prenatal findings is provided in Appendix A.

The patient was born at 39.6 weeks of gestation with a low birth weight (2050 g, −3.0 SD) and height of 44 cm (−2.8 SD), and his head circumference was 27 cm (−3.6 SD). The Apgar scores were recorded as 5/5/6 at 1, 5, and 10 min, respectively. At birth, the patient exhibited moderate asphyxia and respiratory distress syndrome grade III. He required immediate respiratory support, including CPAP and mechanical ventilation. He was admitted to the Neonatal Intensive Care Unit (NICU) immediately after birth due to severe respiratory distress and multiple congenital malformations. The newborn presented in critical condition, exhibiting features of severe intrauterine growth restriction and dysmorphic characteristics. Physical examination revealed microcephaly and facial dysmorphism, including hypotelorism, a high forehead with small palpebral fissures, and a broad nasal bridge (Figure 1A). In addition, a cleft palate (Figure 1B) and bilateral preauricular fistulas (Figure 1C) were observed. Limb anomalies included the absence of the right thumb, hypoplasia of the left thumb, and varus deformity of the left foot (Figure 1D,E). Genital examination showed a concealed and possibly curved phallus, with penoscrotal hypospadias, hypoplasia of the right testis and scrotum, and right-sided cryptorchidism (Figure 1F).

Based on instrumental assessments, the patient demonstrated a functionally single left ventricle with tricuspid valve atresia and a double outlet of the great vessels. Cardiac contractility was preserved, with interatrial communication and a patent ductus arteriosus present. The pericardium appeared intact. A bicuspid aortic valve and severe pulmonary artery hypoplasia were suspected, requiring further evaluation. The pulmonary findings included diffuse reduction in lung aeration, particularly pronounced in the right lung, as well as poorly structured and enlarged pulmonary roots. The diaphragm was smooth, and both costophrenic angles were obscured, consistent with bilateral pleural effusion. Bilateral renal hypoplasia was noted. Central nervous system imaging confirmed alobar holoprosencephaly, consistent with a severe malformation of forebrain development. During early postnatal care, the patient developed bilateral pleural effusion. Diagnostic evaluation of the pleural fluid confirmed chylothorax.

On the second day of life, a dried blood spot sample was sent to the first-tier PCR for TREC and KREC detection. The analysis showed a reduced KREC level (133 copies/10^5^ cells). At 20 days of age, second-tier PCR testing indicated a TREC level of 152 copies/10^5^ cells, with a normal KREC level of 514 copies/10^5^ cells. Because of the syndromic phenotype, the newborn was referred to subsequent diagnostics. Flow cytometry revealed reduced T cell and diminished B cell counts. Notably, an adequate CD4/CD8 ratio and a predominance of naïve T cells were observed (Table 1). Serum immunoglobulin levels were not assessed during the clinical course.

Due to multiple congenital anomalies, the patient was referred for genetic testing, and long-read third-generation sequencing revealed a large deletion:
seq[GRCh38] del(13)(q31.2)dnNC_000013.11:g.88512689_(qter)del
with an estimated size of 25.8 Mb on the long arm of chromosome 13 (Figure 2).

Multiple lines of evidence from the long-read sequencing data support the presence and extent of this deletion: (i) read depth analysis revealed a sharp drop in coverage from an average of 38× upstream of the deletion breakpoint to 18× within the deleted region (chr13:88512689_qter); (ii) a continuous region of autozygosity was detected immediately downstream of the breakpoint, extending 25.8 Mb to the terminal end of chromosome 13; (iii) breakpoint localization placed the deletion start within a long intergenic region between *SLITRK5* and *GPC5*. This breakpoint is supported by 9 out of 31 reads that show a direct transition to the telomeric (GGGTTA)_n_ repeat sequence starting immediately after position chr13:88512688—indicative of a telomere directly joined to this position and confirming the loss of distal chromosomal material. In addition, the long-read sequencing confirmed that the deletion is present in the heterozygous state. It should be noted that no other potential causes of immunodeficiency were revealed.

To confirm the identified deletion, FISH analysis was performed using a DNA probe specific to the subtelomeric region of the long arm of chromosome 13. It revealed a single copy of the SHGC-145617 locus in the 13q34 region in all the analyzed cells, thus validating the sequencing result (Figure 3).

Cytogenetic study of the parents revealed a normal karyotype, suggesting the deletion had occurred de novo in the proband.

The clinical, imaging, laboratory, and genetic findings indicate a complex congenital syndrome characterized by multisystem involvement, including severe cardiac malformations, central nervous system anomalies, renal hypoplasia, dysmorphic features, and early-onset lymphopenia with reduced T and B cell counts.

The patient experienced a complicated infectious course, including pneumonia, pleuritis, and chylothorax, which were managed with antibiotic therapy and surgical interventions, yet his condition progressively deteriorated due to the severity of the congenital defects. The patient died at the age of 7 months.

## 4. Discussion

Chromosome 13 is the largest acrocentric chromosome, characterized by a low gene density of approximately 6.5 genes per megabase [17]. 13q deletion syndrome is a rare chromosomal syndrome resulting from a partial deletion of the long arm of chromosome 13. About 150 patients have been described previously [18]. The clinical presentation of 13q deletion syndrome is influenced by the size and specific location of the deleted region [19,20,21]. Common clinical features include moderate to severe intellectual disability, psychomotor delay, muscular hypotonia, seizures, growth delay, craniofacial dysmorphism, and various congenital defects affecting the brain, eyes, gastrointestinal tract, urogenital system, kidneys, and heart [19,22,23,24,25]. The syndrome may arise de novo or as an inherited condition due to parental chromosomal aberrations [26].

The correlation between the phenotype and genomic aberration in patients having 13q deletion syndrome is not well-established in the literature, largely due to the variability in the sizes of the deleted segments and the challenges in accurately mapping chromosomal breakpoints. Early studies by Brown et al. proposed a classification of three groups based on whether the critical 13q32 band was involved in the deletion [19]. The groups were (i) proximal deletions with an intact 13q32 band, linked to mild intellectual disability, growth delay, and variable retinoblastoma occurrence; (ii) deletions including the 13q32 band, associated with severe congenital malformations; and (iii) distal deletions sparing 13q32, resulting in severe intellectual disability without brain malformations or growth delay [19]. In our case, long-read third-generation sequencing refined the chromosomal abnormality as a 25.8 Mb terminal deletion of 13q, extending from chr13:88,512,689 to qter. Unlike conventional karyotyping or chromosomal microarray analysis (CMA), which would have only indicated the presence of a large terminal 13q deletion, the long-read sequencing allowed us to precisely define the breakpoints at nucleotide resolution and identify the exact gene content of the deleted segment. The long-read sequencing also revealed the presence of a telomeric repeat directly adjacent to the deletion breakpoint, supporting that the loss of the 13q31.2–qter segment did not result from alternative cytogenetic events (e.g., an unbalanced translocation). This level of resolution provides critical insights into the gene content and enables more accurate genotype–phenotype correlations, facilitating comparison with previously reported cases.

The deleted segment encompasses more than 100 genes, including several having high pLI scores (≥0.9) and documented haploinsufficiency (HI) sensitivity, as annotated in ClinGen, DECIPHER, and gnomAD. Particularly noteworthy are genes having autosomal dominant inheritance associated with multisystem developmental phenotypes.

The most prominent among these is *ZIC2* (Zinc Finger of the Cerebellum 2), located at 13q32, indicating a high likelihood of pathogenicity upon deletion. *ZIC2* plays a key role in early neural tube and forebrain development. Heterozygous loss-of-function mutations in *ZIC2* are known to cause holoprosencephaly type 5 (HPE5) (OMIM: 603073) [27,28], characterized by the failure of midline brain structures to properly form [29]. The alobar holoprosencephaly observed in our patient is consistent with complete *ZIC2* haploinsufficiency, making it a likely major contributor to the CNS phenotype.

Another critical gene in the deleted region is *EFNB2* (13q33), encoding ephrin-B2, a membrane-bound ligand essential for vascular morphogenesis, cell–cell signaling, and urogenital development. A human case of *EFNB2* haploinsufficiency has been reported with overlapping features of distal 13q deletions, including anorectal malformations, cardiac anomalies, and genitourinary defects [30]. In our patient, the presence of penoscrotal hypospadias and bilateral renal hypoplasia supports a contributory role for *EFNB2* loss.

The *COL4A1* and *COL4A2* genes (type IV collagen alpha chains), located at 13q34, are both included in the deleted segment. They are highly intolerant to loss of function and critical for the structural integrity of basement membranes in the vasculature and brain. Pathogenic variants cause cerebral small vessel disease, porencephaly, and ocular anomalies [31]. Furthermore, heterozygous loss or mutations in *COL4A1/2* are associated with perinatal intracerebral hemorrhage, porencephalic cysts, ocular dysgenesis, renal involvement, and myopathy, with clinical expression showing marked variability and age dependency [31,32,33]. While many manifestations typically emerge later in life, neonatal cases with severe intracranial hemorrhage and structural brain anomalies have also been described [34]. Therefore, even though our patient did not survive long enough to display late-onset vascular complications, hemizygous deletion of *COL4A1/2* could plausibly have contributed to increased cerebral vulnerability and to the severity of the CNS phenotype observed. Their loss remains relevant, especially in light of the brain and eye involvement commonly reported in 13q deletion syndrome.

Two additional glypican genes—*GPC5* and *GPC6*, involved in skeletal development and growth regulation—are located near the proximal breakpoint and are encompassed by the deletion. Although most pathogenic variants in *GPC6* and *GPC5* are linked to autosomal recessive skeletal dysplasias, emerging evidence suggests that haploinsufficiency may also contribute to skeletal anomalies through dose-dependent effects. In mice, Gpc6^+/−^ heterozygotes show significantly shortened long bones compared to wild-type mice, supporting a dosage-sensitive role in skeletal development [35]. Clinically, a hypomorphic homozygous *GPC6* variant has been associated with milder skeletal dysplasia, including short stature and rhizomelia, indicating that partial loss of function can be pathogenic [36]. For *GPC5*, rare hemizygous deletions have been observed in individuals having growth and digital anomalies, though interpretation is limited due to concurrent deletion of nearby regulatory regions [37], which is in line with the growth delay and limb anomalies observed in our case.

Deletions involving the distal long arm of chromosome 13 (notably 13q33–q34 and overlapping 13q31.2–qter intervals) are well documented to cause a multisystem phenotypic spectrum including developmental delay, dysmorphic features, CNS malformations, congenital heart defects, and urogenital anomalies. Aggregated data from a recent systematic review of 13q33–q34 microdeletions (*n* = 60) reported developmental delay in ~82% of cases, facial dysmorphism in ~50%, seizures in ~20%, and cardiac anomalies, short stature, and hypotonia each in ~10–30% of patients [38]. Earlier series and reports also emphasize frequent male genital/urogenital anomalies mapped to the distal 13q region [25,39], whereas renal anomalies are described more sporadically in individual case reports [40]. Importantly, comprehensive literature and database surveys do not identify immunological dysfunction such as neonatal lymphopenia or abnormal TREC/KREC screening as a reproducible feature of isolated distal 13q deletions. Therefore, the lymphopenia observed in our patient represents a novel or rarely reported observation in the context of 13q31.2–qter loss and warrants consideration of both secondary causes and potential primary genetic contributors within the deleted interval.

On the one hand, chromosomal syndromes are increasingly recognized as potential causes of primary immune dysfunction, even in the absence of deletions involving established IEI-associated genes [9,41,42]. Reported mechanisms include direct disruption of immune-related genes, increased genomic instability leading to lymphocyte apoptosis, and perturbations of chromatin organization or epigenetic regulation [43,44,45,46]. These observations provide a rationale to consider whether the profound lymphopenia in our patient may reflect a primary genetic consequence of the 13q31.2–qter deletion. It is also worth noting that the deleted 13q31.2–qter interval in our patient encompasses more than 100 annotated genes, several of which could plausibly influence immune development or lymphocyte homeostasis. The most compelling candidate is *LIG4* (13q33–q34), encoding DNA ligase IV, a key enzyme of the non-homologous end-joining pathway essential for repair of DNA double-strand breaks and for V(D)J recombination during lymphocyte development. Biallelic pathogenic variants in *LIG4* cause the well-described *LIG4* deficiency syndrome with microcephaly, growth retardation, radiosensitivity, and combined immunodeficiency [47]. Although severe immunodeficiency is typically documented in homozygous or compound heterozygous states, recent evidence suggests that even monoallelic disruption of *LIG4* can contribute to immune dysregulation [48]. Thus, haploinsufficiency of *LIG4* in terminal 13q deletions may plausibly exacerbate defects in lymphocyte maturation and survival. Another important candidate is *MIR17HG* (miR-17~92 cluster) at 13q31.3, which regulates lymphocyte proliferation and survival. In experimental models, deletion of this cluster impairs B-cell development at the pro- to pre-B transition and alters T-cell and plasma-cell responses [49,50,51]. Germline deletions of *MIR17HG* have also been reported in humans (Feingold syndrome) [37], although reproducible immunological phenotypes in carriers remain limited. A further gene of interest within the deleted interval is *EFNB2*, encoding ephrin-B2. Beyond its developmental roles, experimental studies indicate that ephrin-B2 is involved in thymic epithelial cell–thymocyte interactions and T-cell differentiation [52,53]. Thus, *EFNB2* haploinsufficiency may not only contribute to the structural anomalies observed in distal 13q deletions but could also affect immune ontogeny, making it a plausible candidate gene for primary immunological involvement in our case. Finally, *COL4A1* and *COL4A2*, encoding type IV collagen chains, are well known for their roles in cerebrovascular and ocular disease. Because collagen IV is an essential component of basement membranes [54], haploinsufficiency could conceivably influence stromal microenvironments, including those supporting thymic architecture, but this remains speculative. Taken together, these candidate genes highlight that primary genetic contributions to immune dysfunction cannot be excluded in distal 13q deletions.

On the other hand, the reduction in T and B cell counts observed in our patient may be interpreted as a secondary phenomenon related to severe perinatal complications, including neonatal pneumonia, bilateral pleuritis, and a documented chylothorax—conditions known to cause transient lymphopenia and immune suppression.

It is important to note that secondary T-cell lymphopenia may be detected in a substantial proportion of infants during TREC-based newborn screening. According to published estimates, secondary causes account for 34–57% of positive TREC screens. These cases are often linked to perinatal stress, prematurity, congenital anomalies, intrauterine growth restriction, or in utero exposure to immunosuppressive drugs [55,56,57,58]. A well-documented cause among these is neonatal chylothorax, which results in loss of lymphocytes and immunoglobulins via the thoracic duct. It may arise from congenital lymphatic malformations, central venous thrombosis, or postoperative complications, such as after cardiac surgery, and is associated with transient T- and B-cell reductions [59]. Recognition of such secondary mechanisms is crucial for interpreting abnormal TREC/KREC results.

Thus, while the immunological findings in our patient are likely multifactorial—shaped by both critical illness and underlying congenital anomalies—a primary immunological contribution cannot be entirely excluded. The presence of a large terminal chromosomal deletion, combined with early-onset lymphopenia detected through newborn screening, underscores the need for further functional studies and long-term follow-up. Identification of additional patients having similar deletions may help refine genotype–phenotype correlations and clarify the potential role of 13q genes in immune development. In our case, early detection of immune abnormalities prompted timely supportive intervention. Although preemptive immunoglobulin therapy did not alter the overall clinical trajectory, it may have contributed to partial infection control and temporary immune stabilization. This experience illustrates the potential benefit of early immunological assessment and management in syndromic neonates, even in the absence of established IEI-related genes.

## 5. Conclusions

In summary, we present a rare and well-characterized case of 13q31.2–qter deletion syndrome, in which we precisely defined the deletion boundaries using state-of-the-art long-read sequencing, a method that offers superior resolution for structural variants compared to traditional short-read methods [60]. Patients having large deletions of 13q—including the terminal 13q31.2 region—are uncommon, with about 150 cases reported, and most affected infants do not survive beyond early infancy.

Our patient exhibited mild lymphopenia during the neonatal period, a finding not previously reported in cases of distal 13q deletion. However, it remains unclear whether this immunological abnormality is primary, resulting from a genetic defect, or secondary to systemic illness—specifically severe infection, inflammation, and chylothorax. Similar immune disturbances have been observed in other syndromic infants, underscoring the need for careful monitoring and, when appropriate, timely immunomodulatory treatment. Clinically, the syndrome was marked by multiorgan involvement—central nervous system malformation, complex congenital heart disease, renal hypoplasia, urogenital anomalies, and pulmonary complications—leading to death at 7 months of age. This outcome is consistent with the known severe phenotype associated with large distal 13q deletions.

Implementing long-read sequencing not only allows accurate localization of breakpoints, but this high-resolution approach also strengthens genotype–phenotype correlations and illustrates how advanced genomic technologies can enhance the diagnostic toolkit for rare chromosomal syndromes.

## Figures and Tables

**Figure 1 ijms-26-09302-f001:**
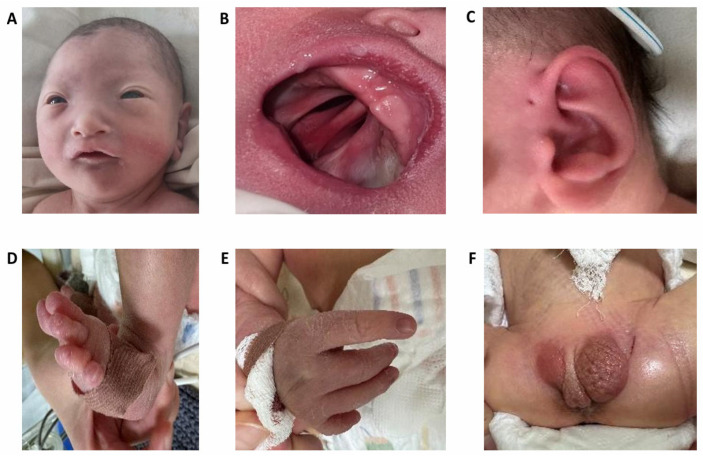
Multiple anomalies in the proband having del13q. (**A**)—facial features; (**B**)—complete cleft palate; (**C**)—preauricular fistula, (**D**)—foot anomalies, (**E**)—hand anomalies, (**F**)—genital anomalies.

**Figure 2 ijms-26-09302-f002:**
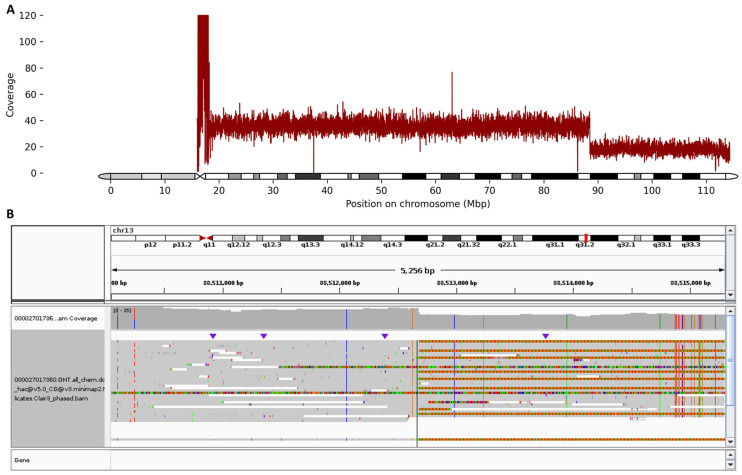
(**A**) Chromosome 13 ideogram and sequencing coverage plot (capped at 120) showing a clear two-fold decrease in read depth at 13q31.2, consistent with a heterozygous terminal deletion. The x-axis represents chromosomal coordinates, while the y-axis indicates normalized sequencing depth across chromosome 13. The sharp drop in coverage delineates the deleted interval. (**B**) Integrative Genomics Viewer (IGV) browser view illustrating the precise breakpoint of the heterozygous 25.8 Mb deletion on the long arm of chromosome 13 in the proband. The immediate start of continuous telomeric repeat is also clearly visualized.

**Figure 3 ijms-26-09302-f003:**
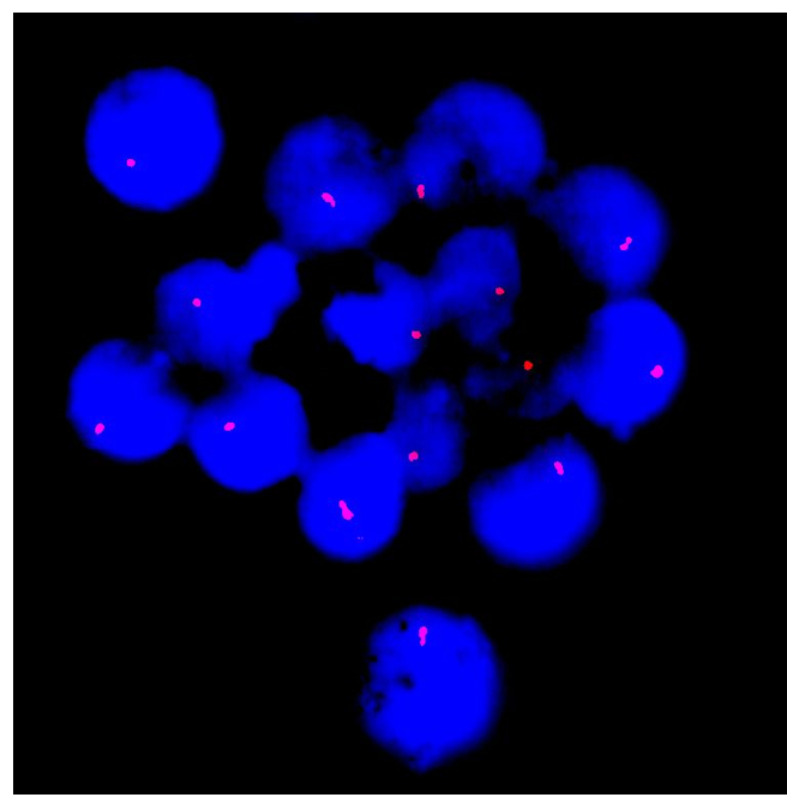
Fluorescence in situ hybridization (FISH) analysis of interphase nuclei (stained with DAPI and seen in blue) with probes specific for the subtelomeric region of chromosome 13q34 (red signals). Each nucleus demonstrates a single signal instead of the expected two, consistent with a heterozygous deletion at 13q34 and supporting the sequencing results.

**Table 1 ijms-26-09302-t001:** Lymphocyte subpopulation analysis by flow cytometry.

Parameter	%	Cells (×10^6^/mL)	Normal %	Normal Cell Count (×10^6^/mL)
White Blood Cells (WBC)	-	7.09	-	6.2–12.1
Granulocytes	59.0	4.18	14.7–35.7	1.0–3.2
Monocytes	15.0	1.06	4.2–11.6	0.3–1.0
Lymphocytes	25.0	1.77	57.5–79.5	4.0–8.0
	**%**	**Cells (×10^9^/L)**	**Normal %**	**Normal cell count (×10^6^/mL)**
T-cells CD3+Lym	76.8	1.361	58.3–73.9	2.7–5.1
CD3+CD4+Lym	70.3	0.957	56.6–79.1	1.7–3.6
T-naïve cells CD4+CD45RA+CD197+	72.3	0.692	73.4–87.5	1.393–2.896
T-central memory cells CD4+CD45RA−CD197+	10.6	0.101	4.8–15.8	0.110–0.499
Effector memory cells CD4+CD45RA−CD197−	11.2	0.107	4.0–12.6	0.107–0.342
TEMRA CD4+CD45RA+CD197−	6.0	0.057	1.0–6.1	0.027–0.169
CD3+CD8+Lym	24.9	0.339	17.9–38.9	0.6–1.8
T-naïve cells CD8+CD45RA+CD197+	93.1	0.316	47.6–85.1	0.381–1.182
T-central memory cells CD8+CD45RA−CD197+	0.2	0.001	0.3–5.6	0.003–0.074
Effector memory cells CD8+CD45RA−CD197−	0.1	0.000	2.6–20.9	0.028–0.281
TEMRA CD8+CD45RA+CD197−	6.7	0.023	6.2–35.9	0.067–0.497
B-cells CD19+Lym	4.5	0.080	18.7–33.3	0.8–2.3
NK-cells CD3−CD16+CD56+Lym	18.7	0.331	3.0–12.9	0.2–0.9
CD56+high NK	16.6	0.055	4.1–19.4	0.01–0.08
T-NK-cells CD3+CD56+Lym	0.4	0.007	0.1–0.6	0.00–0.03

## Data Availability

The data that support the findings of this study are available on request from the corresponding author. The data are not publicly available due to privacy or ethical restrictions.

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
