# Peer review of "13q Deletion Syndrome Presenting with Lymphopenia Detected Through Newborn Screening for Primary Immunodeficiencies"

_ijms, 2025, doi:10.3390/ijms26199302_

Round 1
Reviewer 1 Report
Comments and Suggestions for Authors
Review comments for Efimova et al. 2025:
General and need of change: how come the Correspondence Email and telephone number belong in Spain, when the main research is in Russia? I hope that is a mistake and needs to be corrected.
Introduction
Line 58 – I believe citation 10 is not proper. I read the quoted publication and I did not reach the same conlusion as the written sentence.
M&M
Line 91 – I tried to search (in Germany) and read citation 14 for the reference values, but I failed to find the full publication in English.
Line 96 – citation to manufacturer’s protocol or short description?
Clinical Report
Very informative and I don’t have any comments.
Discussion
no comments

Author Response
We sincerely thank the Reviewer for the careful reading of our manuscript and for the constructive comments and suggestions provided. The feedback has been very helpful in improving both the clarity and scientific rigor of the paper. We have addressed all points raised and believe that the revised version has benefited greatly from these remarks.
All line numbers mentioned in our responses refer to the revised version of the manuscript with accepted changes (“ijms-3855808-accepted changes”).
- General and need of change: how come the Correspondence Email and telephone number belong in Spain, when the main research is in Russia? I hope that is a mistake and needs to be corrected.
Response: We thank the reviewer for this comment. However, Spain is not mentioned anywhere in the manuscript, including in the affiliations or correspondence details. All institutional affiliations belong to Russia, and the e-mail addresses provided are either institutional accounts or international e-mail services. We believe this may be a misunderstanding.
- Line 58 – Citation 10 appears to be inappropriate. Upon reviewing the quoted manuscript, I
was unable to establish a clear connection between the authors’ statements and the cited work. It also seems that the authors frequently cite their own publications; I counted at least six self-citations. Is there a specific justification for this?
Response: Thank you for this important point. We believe that Citation 10 (PMID: 39409174) is appropriate, because in that work we demonstrated that TREC/KREC-based newborn screening identifies not only classical SCID / severe T- or B-cell deficiencies but also syndromic conditions with immune abnormalities, some of which fall outside the currently established IEI classifications. In addition, we have searched for and added several independent studies showing that TREC/KREC newborn screening picks up syndromic conditions or chromosomal anomalies with associated immune abnormalities. These studies strengthen the statement that non-SCID immune defects are identifiable. We have updated the manuscript sentence accordingly to reflect these additional evidences. (line 56 - 59)
Regarding the number of self-citations, we have reviewed them: we retained those that are directly relevant (methods, prior observations foundational to this work), and replaced or supplemented others with independent literature wherever possible, to strengthen objectivity.
- Line 91 – I attempted to access citation 14 to verify the reference values mentioned, but was
unable to locate a full English version of the publication (at least in Germany). The authors
might consider including the reference values as supplementary material to facilitate verification and a better understanding
Response: We fully acknowledge that Reference 14 is a Russian-language publication, which may not be readily accessible to an international audience. To address this concern, we have included the relevant reference values for lymphocyte subpopulations directly in the manuscript (see Table 1). This ensures that all readers can verify and better understand the immunological findings without the need to access the original Russian source.
- Line 96 – A citation or brief description of the manufacturer’s protocol is needed. I was unable to find a source for the protocol in Germany, so adding this information would help understanding and comparison to other articles.
Response: We appreciate this important point. We have now added a citation to the manufacturer’s protocol for the Oxford Nanopore Ligation Sequencing Kit (SQK-LSK114) in the Materials and Methods section, including the link to the official documentation, to facilitate reproducibility and comparison with other studies. (lines 95 - 108)

Reviewer 2 Report
Comments and Suggestions for Authors
13q Deletion Syndrome Presenting with Lymphopenia Detected through Newborn Screening for Primary Immunodeficiencies
By Efimova et. al.
General Assessment
This is a well-prepared and detailed case report that describes a rare instance of 13q deletion syndrome detected through newborn screening for immunodeficiencies. The combination of clinical, immunological, and genomic findings, along with the use of long-read sequencing and confirmatory FISH, makes the case compelling and highly relevant for the journal’s readership. The manuscript is clearly written, generally well-structured, and contributes valuable information to the literature.
At the same time, there are areas where the discussion could be further developed, the presentation clarified, and some details refined. Below, I provide major and minor comments to help strengthen the work.
Major Comments
- Primary vs. Secondary Lymphopenia: The case highlights lymphopenia, which has not been consistently reported in 13q deletion syndrome. The discussion touches on whether this is primary (a direct result of the deletion) or secondary (due to infections, chylothorax, or perinatal stress), but the argument remains somewhat speculative. I recommend expanding on genes within 13q that might plausibly affect immune function, and more clearly distinguishing what evidence supports one explanation over the other.
- Serum immunoglobulin levels were not assessed, which is a limitation. Please comment explicitly on how this absence impacts interpretation. Could hypogammaglobulinemia have been present? Would this have changed management?
- Flow cytometry results (Table 1) are detailed, but the clinical significance would be clearer if the authors highlighted in the text which cell populations were most abnormal compared to age-matched references.
- The use of long-read sequencing is an important strength. However, the discussion of the genes within the deleted interval could be expanded. For example, COL4A1 and COL4A2 are mentioned only briefly, despite their strong disease associations. Please elaborate on how their deletion might have contributed to the phenotype, even if not clinically evident within the short lifespan.
- The legends for Figures 2-3 are very brief. Please expand them to guide readers who may not be familiar with sequencing read depth plots or FISH interpretation.
- The discussion cites the classic Brown et al. classification but could go further in situating this patient among the ~150 reported cases of distal 13q deletions. How common are heart, renal, and urogenital anomalies in this deletion interval? Are there any reports in DECIPHER or the literature of immune involvement in similar cases? Strengthening this comparison will highlight the novelty of your case.
- When emphasizing the utility of long-read sequencing, it would be useful to clarify what would have been missed or less precisely characterized using only conventional methods such as CMA or karyotyping.
- The timeline of anomalies detected during pregnancy is somewhat difficult to follow. It would be clearer if the findings were presented in a chronological sequence by trimester or summarized in a supplementary table. This would help clinicians quickly grasp the prenatal progression.
Minor Comments
- The abstract is somewhat dense. Consider shortening long sentences to improve readability.
- It would be worth explicitly noting that this is the first reported case of 13q deletion syndrome identified via newborn screening.
- Figure 1 (clinical photographs) should be clearly labeled with arrows to indicate the anomalies, as not all are immediately obvious. Ensure the resolution and anonymization meet journal standards.
- Presentation of Figure 1 could also benefit from considering or maintaining a similar size for all the panels
- Figures 2 and 3 need more descriptive legends to make them understandable to non-specialist readers.
- The text alternates between “primary immunodeficiencies (PID)” and “inborn errors of immunity (IEI).” Please standardize terminology according to the latest IUIS classification.
- Some references (e.g., 28-30) feel tangential to the main discussion. Consider revising these and including more directly relevant citations on 13q syndrome and immune manifestations in chromosomal disorders.
- The manuscript includes a statement on ChatGPT use. While acceptable under MDPI policy, it may be more appropriate to place this in the Acknowledgments section rather than as a standalone declaration. (If/ unless recommended by the Journal)
The manuscript represents a valuable contribution. With expanded discussion of the immunological findings, deeper integration into the literature, clearer figure legends, and some editorial refinements, it will be suitable for publication. Hence, being a case report, I would like to suggest a Minor Revision
Author Response
We would like to express our gratitude to the Reviewer for the thorough evaluation and valuable feedback. The remarks have helped us to refine the manuscript, and we are confident that the revised version has been significantly improved as a result.
All line numbers mentioned in our responses refer to the revised version of the manuscript with accepted changes (“ijms-3855808-accepted changes”).
Major Comments
- Primary vs. Secondary Lymphopenia: The case highlights lymphopenia, which has not been consistently reported in 13q deletion syndrome. The discussion touches on whether this is primary (a direct result of the deletion) or secondary (due to infections, chylothorax, or perinatal stress), but the argument remains somewhat speculative. I recommend expanding on genes within 13q that might plausibly affect immune function, and more clearly distinguishing what evidence supports one explanation over the other.
Response: We appreciate the reviewer’s thoughtful comment. In the revised manuscript, we refined the section addressing chromosomal aberrations and their general impact on immunity, and expanded the discussion of specific candidate genes within the deleted 13q31.2–qter interval that could plausibly contribute to primary immunodeficiency (see Lines 306–341 in the revised manuscript, file “ijms-3855808-accepted changes”).
In Lines 306–341, we discuss both general mechanisms by which chromosomal deletions may predispose to immune alterations and highlight several candidate genes within the deleted region that may be directly relevant to lymphocyte development and function, thus providing arguments in favor of a possible primary contribution to the observed lymphopenia.
In Lines 342 - 355, we outline alternative explanations supporting secondary causes of lymphopenia, including infection, chylothorax, and perinatal stress.
By presenting these two perspectives in separate sections, we aimed to more clearly distinguish the evidence for primary versus secondary mechanisms and emphasize that both remain plausible in this case.
- Serum immunoglobulin levels were not assessed, which is a limitation. Please comment explicitly on how this absence impacts interpretation. Could hypogammaglobulinemia have been present? Would this have changed management?
Response: We agree that the absence of immunoglobulins is a limitation of this case study.
The syndromic phenotype, low TREC or KREC levels, and decreased T or B lymphocytes counts warrant a combined PID and IVIG therapy approach, regardless of immunoglobulin levels. Immunoglobulin levels are not the only parameter used to determine IVIG therapy indications due to the numerous potential causes of hypogammaglobulinemia in children with syndromic pathology and infectious complications (e.g., physiological hypogammaglobulinemia, chylothorax etc).
- Flow cytometry results (Table 1) are detailed, but the clinical significance would be clearer if the authors highlighted in the text which cell populations were most abnormal compared to age-matched references (Lines 288–325).
Response: We thank the reviewer for this helpful suggestion. In the revised manuscript, we have clarified the clinical significance of the flow cytometry findings (Lines – 168-170).
- The use of long-read sequencing is an important strength. However, the discussion of the genes within the deleted interval could be expanded. For example, COL4A1 and COL4A2 are mentioned only briefly, despite their strong disease associations. Please elaborate on how their deletion might have contributed to the phenotype, even if not clinically evident within the short lifespan.
Response: We thank the reviewer for this valuable suggestion. In the revised manuscript, we have expanded the discussion of COL4A1 and COL4A2 (lines 268 - 277). At the same time, we would like to mention that while we highlighted several major candidate genes within the deleted interval, a comprehensive review of all affected genes was beyond the scope of the present study.
- The legends for Figures 2-3 are very brief. Please expand them to guide readers who may not be familiar with sequencing read depth plots or FISH interpretation.
Response: We are grateful for this constructive feedback. The legends for Figures 2 and 3 have been expanded.
- The discussion cites the classic Brown et al.classification but could go further in situating this patient among the ~150 reported cases of distal 13q deletions. How common are heart, renal, and urogenital anomalies in this deletion interval? Are there any reports in DECIPHER or the literature of immune involvement in similar cases? Strengthening this comparison will highlight the novelty of your case.
Response: We carefully reviewed the literature and relevant databases to place our patient in the context of previously reported distal 13q deletions. We also highlighted the relative frequency of cardiac, renal, and urogenital anomalies, and emphasized the absence of immune involvement in similar cases. These points have now been incorporated into the revised Discussion (lines 291–305).
- When emphasizing the utility of long-read sequencing, it would be useful to clarify what would have been missed or less precisely characterized using only conventional methods such as CMA or karyotyping.
Response: We thank the reviewer for this important comment. In the revised manuscript, we have added a clarification. (Lines 238-246)
- The timeline of anomalies detected during pregnancy is somewhat difficult to follow. It would be clearer if the findings were presented in a chronological sequence by trimester or summarized in a supplementary table. This would help clinicians quickly grasp the prenatal progression.
Response: We thank the reviewer for this constructive suggestion. In the revised manuscript, we have reorganized the description of prenatal findings according to trimesters and provided this information in a supplementary table (Table S1) summarizing the anomalies detected during pregnancy. We agree that this format improves clarity and allows clinicians to better follow the prenatal progression.
Table S1. Chronological summary of prenatal findings by trimester in the proband with 13q deletion syndrome.
|
Trimester |
Gestational age |
Findings |
|
First trimester |
3–4 weeks |
Threatened miscarriage; hospitalization for pregnancy preservation; maternal iron-deficiency anemia |
|
~12 weeks |
Increased risk of preeclampsia and fetal growth restriction (FGR); multiple congenital anomalies suspected; high risk of chromosomal abnormalities; prenatal karyotyping was recommended but not performed |
|
|
Second trimester |
16–18 weeks |
Maternal iron-deficiency anemia, hypothyroxinemia; suspected CNS anomalies; fetal growth restriction |
|
19 weeks |
Confirmed FGR and multiple anomalies: CNS, congenital heart defect, urogenital malformations; high risk for chromosomal abnormalities confirmed by perinatal consultation |
|
|
Third trimester |
29 weeks |
Alobar holoprosencephaly; functionally single left ventricle with tricuspid valve atresia; double outlet of the great vessels with severe pulmonary artery hypoplasia/atresia; bilateral renal hypoplasia; progressive FGR |
|
39 weeks |
Severe growth restriction (birth weight 2050 g); alobar holoprosencephaly, congenital heart defect (single ventricle, tricuspid atresia, great vessel anomaly), renal hypoplasia; oligohydramnios; orofacial cleft suspected |
Minor Comments
1. The abstract is somewhat dense. Consider shortening long sentences to improve readability.
Response: We thank the reviewer for this suggestion. The abstract has been revised, and long sentences were shortened to improve clarity and readability.
2. It would be worth explicitly noting that this is the first reported case of 13q deletion syndrome identified via newborn screening.
Response: Thank you for this comment. In our case, a structural chromosomal abnormality was already suspected prenatally, so the diagnosis of 13q deletion syndrome would have been established independently of newborn screening. However, the newborn screening program revealed lymphopenia, which prompted detailed immunological investigations as well as confirmatory cytogenetic and molecular studies (including FISH and long-read sequencing).
3. Figure 1 (clinical photographs) should be clearly labeled with arrows to indicate the anomalies, as not all are immediately obvious. Ensure the resolution and anonymization meet journal standards.
Response: We appreciate this comment. The resolution and anonymization of Figure 1 have been checked and adjusted in line with journal requirements. However, we did not add arrows, since most anomalies represent overall morphological features (such as dysmorphic facial characteristics, cleft palate, limb malformations, and genital anomalies), which are better appreciated in their entirety rather than by highlighting single points. We believe that arrows would not significantly improve clarity in this case.
4. Presentation of Figure 1 could also benefit from considering or maintaining a similar size for all the panels
Response: We thank the reviewer for this suggestion. While the panels could not be made exactly the same size due to differences in image proportions, we have improved the layout of Figure 1 to achieve a more balanced appearance and better overall clarity.
5. Figures 2 and 3 need more descriptive legends to make them understandable to non-specialist readers.
Response: Thank you for pointing this out. We expanded and clarified the figure legends for Figures 2 and 3 to make them more informative and accessible for non-specialist readers.
6. The text alternates between “primary immunodeficiencies (PID)” and “inborn errors of immunity (IEI).” Please standardize terminology according to the latest IUIS classification.
Response: We appreciate this important comment. In accordance with the latest IUIS classification, we have standardized the terminology in the manuscript to “inborn errors of immunity (IEI)”. Since the term “primary immunodeficiencies (PID)” remains in common clinical use, particularly in the context of newborn screening programs, we have kept it in the title.
7. Some references (e.g., 28-30) feel tangential to the main discussion. Consider revising these and including more directly relevant citations on 13q syndrome and immune manifestations in chromosomal disorders.
Thank you for pointing this out. We have revised the EFNB2 section and replaced the previous references with directly relevant studies (lines 258 - 263).
8. The manuscript includes a statement on ChatGPT use. While acceptable under MDPI policy, it may be more appropriate to place this in the Acknowledgments section rather than as a standalone declaration. (If/ unless recommended by the Journal).
Response: We thank the reviewer for this excellent suggestion. We have moved the statement regarding the use of ChatGPT into the Acknowledgments section. (lines 407- 409).
